# Minibody-Based and scFv-Based Antibody Fragment-Drug Conjugates Selectively Eliminate GD2-Positive Tumor Cells

**DOI:** 10.3390/ijms24021239

**Published:** 2023-01-08

**Authors:** Daniel V. Kalinovsky, Irina V. Kholodenko, Alexey V. Kibardin, Igor I. Doronin, Elena V. Svirshchevskaya, Dmitriy Y. Ryazantsev, Maria V. Konovalova, Fedor N. Rozov, Sergey S. Larin, Sergey M. Deyev, Roman V. Kholodenko

**Affiliations:** 1Shemyakin-Ovchinnikov Institute of Bioorganic Chemistry, Russian Academy of Sciences, 117997 Moscow, Russia; 2Orekhovich Institute of Biomedical Chemistry, 119121 Moscow, Russia; 3Dmitry Rogachev Federal Research Center of Pediatric Hematology, Oncology and Immunology, 117997 Moscow, Russia; 4Real Target LLC, 108841 Moscow, Russia; 5Lomonosov Moscow State University, 119991 Moscow, Russia; 6Sechenov First Moscow State Medical University, 119992 Moscow, Russia

**Keywords:** antibody fragments, scFv fragments, minibodies, antibody-drug conjugates, ganglioside GD2, immunotherapy, cancer, GD2-positive tumors, neuroblastoma, melanoma

## Abstract

Ganglioside GD2 is a well-established target expressed on multiple solid tumors, many of which are characterized by low treatment efficiency. Antibody-drug conjugates (ADCs) have demonstrated marked success in a number of solid tumors, and GD2-directed drug conjugates may also hold strong therapeutic potential. In a recent study, we showed that ADCs based on the approved antibody dinutuximab and the drugs monomethyl auristatin E (MMAE) or F (MMAF) manifested potent and selective cytotoxicity in a panel of tumor cell lines and strongly inhibited solid tumor growth in GD2-positive mouse cancer models. Here, we employed two different GD2-binding moieties–minibodies and scFv fragments that carry variable antibody domains identical to those of dinutuximab, and site-directly conjugated them to MMAE or MMAF by thiol-maleimide chemistry with drug-to-antibody ratios (DAR) of 2 and 1, respectively. Specific binding of the antibody fragment-drug conjugates (FDCs) to GD2 was confirmed in direct ELISA, flow cytometry, and confocal microscopy. Selective cytotoxic and cytostatic effects of the conjugates were observed in GD2-positive but not GD2-negative neuroblastoma and melanoma cell lines. Minibody-based FDCs demonstrated more pronounced cytotoxic effects and stronger antigen binding compared to scFv-based FDCs. The developed molecules may offer considerable practical benefit, since antibody fragment-drug conjugates are capable of enhancing therapeutic efficacy of ADCs by improving their pharmacokinetic characteristics and reducing side effects.

## 1. Introduction

Ganglioside GD2 is a validated clinical target overexpressed in multiple cancers such as neuroblastoma, glioma, breast cancer, sarcomas, small cell lung cancer, and melanoma. Its postnatal expression on healthy body cells is generally limited to the central nervous system, peripheral neurons, and skin melanocytes. Ganglioside GD2 performs complex biological functions in normal cells, including participation in cellular communication and adhesion, formation of lipid rafts, and modulation of signaling pathways. Being a glycosphingolipid, GD2 is directly synthesized from its precursor ganglioside GD3 by the enzyme GM2/GD2 synthase; however, its content in the cell is determined by the combined action of the various enzymes involved in ganglioside biosynthesis [1]. The most significant changes in GD2 metabolism occur during malignant transformation of cells, making it an attractive marker for antitumor therapy.

The monoclonal antibodies dinutuximab and naxitamab, chimeric and humanized antibodies from the 14.18/14G2a and 3F8 families, respectively, are the only GD2-directed therapeutics that have been granted regulatory approval. Extensive research is underway to develop alternative GD2-specific strategies, including bispecific antibodies, immunocytokines, CAR T and NK cell therapies, and radioimmunotherapy [2,3]; however, the rapidly developing area of antibody-drug conjugates (ADCs) has been largely neglected in this regard. The promise of ADCs directed to GD2, which is primarily a marker of solid tumors, is based on the fact that antibody conjugates carrying different cytotoxic payloads have recently shown clinical benefit in a number of solid tumors, with five ADCs reaching approval as solid tumor therapies. Today, over 80% of ADCs in active clinical trials are investigated for solid tumor therapy, a case very different from just ten years ago. Importantly, numerous clinical studies explore ADCs targeting alternative tumor-associated molecules in GD2-expressing tumors including neuroblastoma, melanoma, and sarcomas, while the clinically approved drug conjugates directed to HER2 and Trop-2 markers in breast cancer are considered a major success story [4,5].

Only two works regarding GD2-targeting conjugates with small cytotoxic molecules are present in the open literature. Lode et al. [6] developed an ADC based on the mouse antibody 14G2a and the DNA-cleaving calicheamicin θ^I^_1_ drug, which was conjugated to the modified lysines of the antibody, and the conjugate was shown to significantly suppress liver metastases in a disseminated mouse model of neuroblastoma. In our recent work [7], ADCs were generated using the chimeric ch14.18 variant of the same antibody and the microtubule-depolymerizing agents monomethyl auristatin E (MMAE) or F (MMAF) that were conjugated to antibody interchain cysteines by thiol-maleimide chemistry. Antibody conjugates with both MMAE and MMAF showed high and selective cytotoxicity in a broad panel of GD2-expressing tumor cell lines, and also strongly inhibited the growth of solid tumors in mouse models of melanoma and lymphoma. These studies indicate the validity of employing ADCs for treating GD2-positive human tumors and support their further optimization.

Changing the antibody format within ADCs from full-length IgG molecules to smaller antibody fragments that results in antibody fragment-drug conjugates (FDCs) represents an important strategy for improving the efficacy of ADC therapy. All major antibody fragment formats such as Fab fragments, minibodies, diabodies, scFvs, and nanobodies, as well as smaller antigen-binding peptides, are being actively explored in preclinical studies as targeted molecules within drug conjugates; however, it is yet to be evaluated whether they represent a viable alternative to full-length mAbs in drug conjugates [8,9]. The primary advantages of antibody fragments over full-length antibodies generally constitute a more efficient penetration into solid tumors attributed to their smaller size, and their ability to interact with tumor-associated antigens that are inaccessible to full-length mAbs due to steric hindrances on the cell surface. For various formats of antibody fragments, it has been demonstrated that they pass through the walls of blood vessels and diffuse into the tumor faster, and are more homogeneously distributed in it compared to full-length antibodies [10,11]. At the same time, the longer circulation time of full-length mAbs in the blood compared to antibody fragments often leads to their higher accumulation in the tumor over prolonged time; therefore, studies that demonstrate that rapid accumulation of FDCs in the tumor compared with ADCs may also be translated into more beneficial antitumor effects in vivo [12] are of crucial importance for advancing FDCs into the clinic. In addition, rapid systemic clearance of FDCs may potentially provide a wider therapeutic window compared with ADCs that require considerable time to accumulate in the tumor and take weeks to be cleared from the body. Finally, the inability of most FDCs to bind Fc receptors prevents Fc-mediated side effects which have been shown for both full-length antibodies (specifically, the antibody-induced allodynia for GD2-specific dinutuximab [13]) and for ADCs [14].

No data exists on FDCs directed to GD2-expressing tumors. In this work, such FDCs were developed for the first time based on two antibody fragment formats: the scFv fragment from the clinically approved antibody ch14.18, and the corresponding minibody, which is the bivalent dimer of the same scFv fragment fused to the hinge region and the CH3 domain of the heavy chain of IgG1 antibody. MMAE and MMAF were site-directly conjugated to cysteines of the minibody hinge region and the terminal cysteine introduced into the scFv structure via a cleavable maleimide linker. We generated the FDCs with specific drug-to-antibody ratios (DAR) and evaluated their stability and binding to GD2 by the enzyme-linked immunosorbent assay (ELISA) and to cells with or without GD2 expression by flow cytometry and confocal microscopy. Furthermore, the cytotoxic effects of the conjugates were analyzed in GD2-positive and GD2-negative tumor cell lines by the MTT and propidium iodide (PI) assays.

## 2. Results

### 2.1. Production of GD2-Specific scFv Fragments and Minibodies

#### 2.1.1. Design and Expression of the Antibody Fragments

Both the scFv antibody fragment and the minibody were designed based on the sequence of the light (VL) and heavy (VH) chain variable domains of the GD2-specific antibody 14.18 reported by Bolesta et al. [15]. The scFv fragment sequence was arranged in the VL-VH orientation, and the commonly used (G_3_S)_4_ linker for construct flexibility and protease resistance, the FLAG octapeptide tag (the sequence DYKDDDDK) for detection, and a C-terminal unpaired cysteine for site-directed thiol-maleimide conjugation were introduced into the sequence (Figure 1A). The minibody 14.18 consists of the variable VL and VH domains connected by the (G_3_S)_4_ linker identical to and in the same orientation as those in the scFv fragment 14.18, with the addition of the IgG1 hinge region, a second GS linker (GGGSSGGGSG), and the IgG1 CH3 constant domain (Figure 1A).

Following transient gene expression in Expi293F cells and purification by protein L chromatography, both the minibody (Figure 1B) and the scFv fragment (Figure 1D) were characterized by high purity and stability when examined under native conditions in size-exclusion chromatography. The minibody was observed as an individual peak, whose retention time is in good agreement with literature data [16], while the scFv fragment was represented by two overlapping peaks.

We also analyzed the scFv fragment and the minibody in 10% polyacrylamide gel electrophoresis (SDS-PAGE) (Figure 1C,E). The observed molecular weight of the scFv fragment under reducing conditions was approx. 30 kDa (Figure 1E, line 2), which roughly corresponds to its theoretical weight of 27 kDa as calculated by the ProtParam tool (expasy.org). Resolving the scFv fragment under non-reducing conditions, besides the monomer band at 30 kDa, yielded an additional smaller band at approx. 60 kDa that corresponds to the scFv dimer, which is explained by the introduction of the C-terminal cysteine in the structure of the scFv fragment (Figure 1E, line 3). These data explain the complex form of the scFv fragment peaks in size-exclusion chromatography which includes both the scFv monomer and its dimer (Figure 1D). The purified minibody formed a band corresponding to the homodimeric protein with an estimated molecular weight of 85 kDa under non-reducing conditions (Figure 1C, line 3), while the theoretical weight of the minibody dimer is 82 kDa. In reducing conditions, the dimer fully dissociated into the monomer and formed a band at approx. 45 kDa (Figure 1C, line 2).

#### 2.1.2. Antigen-Binding Properties of the Antibody Fragments

Several methods were used to confirm the specific binding of the obtained antibody fragments to the antigen. First, conjugates of the scFv 14.18 and minibody 14.18 with fluorescein 6-maleimide (FAM) were generated, and with these conjugates we directly demonstrated in flow cytometry and in confocal microscopy that both the scFv fragment and the minibody did not bind the GD2-negative B16 melanoma cell line, but manifested high binding to the B78-D14 melanoma cell line overexpressing GD2 (Figure 2A,B). The GD2 status of these cell lines was additionally confirmed by microscopy experiments with the FAM-labeled full-length ch14.18 antibody (Figure 2A). In flow cytometry experiments, the relative fluorescence intensity (RFI) values of B16 cells stained with FAM-labeled antibody fragments constituted 1 ± 0.3 for both antibody fragments, while the RFI of B78-D14 cells stained with minibodies constituted 23.6 ± 5, which was significantly higher than the RFI of 12.2 ± 4 for the scFv fragments (Figure 2B). Higher RFI values for minibodies obtained by flow cytometry analysis correlated with markedly higher fluorescence intensity of cells stained with FAM-labeled minibodies compared to FAM-labeled scFv fragments in confocal microscopy (Figure 2A).

Additionally, the specificity of binding of the obtained antibody fragments was studied in direct ELISA using individual purified gangliosides GD2, GD3, GD1b, and GM2 (Figure 2C). When ganglioside GD2 was sorbed on the surface of the plate wells, a high intensity of the peroxidase reaction was observed, while the sorption of other gangliosides led to very low 3,3′,5,5′-Tetramethylbenzidine (TMB) staining with optical density (OD) values comparable to those of the negative control. The percent of cross-reactivity did not exceed 5% within the concentration interval used to evaluate scFv and minibody binding (up to 50 nM), which indicates a high specificity of interaction of our antibody fragments with ganglioside GD2 (Figure 2C).

### 2.2. Production and Antigen-Binding Properties of the FDCs

We generated the FDCs by conjugating MMAE or MMAF microtubule inhibitors to the C-terminal cysteine of the scFv fragment 14.18 or the reduced interchain cysteines of the minibody 14.18 by a maleimide linker that contains the cathepsin-cleavable valine-citrulline site. The reaction scheme and the FDC structures are presented in Figure 3A. The reaction conditions (primarily the auristatin drug excess) for producing the scFv drug conjugates were optimized to saturate the C-terminal cysteine on the scFv molecule with the drug, resulting in a DAR 1. For the minibody-based conjugates, conditions were chosen that favored the highest DAR and at the same time would ensure stability of the FDC during the reaction and in the subsequent analysis.

Based on the ultraviolet-visible (UV-VIS) spectroscopy analysis, the average drug-to-antibody ratios for both the scFv-MMAE and scFv-MMAF generated in optimized reaction conditions constituted 0.9 and were highly consistent between batches (*n* = 4), while the average DAR for minibody-MMAE and minibody-MMAF in optimized conditions constituted 2.0 ± 0.1 and 2.1 ± 0.1, respectively. The absorption spectra of the representative scFv-based and minibody-based FDCs normalized at 280 nm together with the corresponding naked antibody fragment are shown in Figure 3B, C. It can be seen that the absorption peak of the MMAE/MMAF observed at approx. 250 nm (Figure 3D) results in an increase in the relative absorbance for the FDCs compared to the original proteins.

Additionally, FAM-labeled scFv fragments and minibodies were generated in reaction conditions identical to the FDCs that were used for cell staining in flow cytometry and confocal microscopy experiments described above, and also served for verification of the average number of drugs conjugated per protein molecule within the FDCs. The average degree of labeling for the FAM-labeled scFv and minibody roughly corresponded to the DAR of the FDCs and constituted approx. 0.8 ± 0.1 fluorophore molecules per scFv fragment and 1.7 ± 0.2 molecules per minibody.

The FDCs were resolved in gel electrophoresis in order to analyze their stability, and essentially no difference was observed in the bands between them and the original antibody fragments (Figure 3E). In our earlier work with the full-length GD2-specific antibody ch14.18 thiol-conjugated to MMAE and MMAF, we were able to identify a slight increase in the molecular weight of the bands corresponding to the light and heavy antibody chains of the ADC relative to the initial antibody in 10% reducing SDS-PAGE [7]. However, such difference could not be observed with scFv-based and minibody-based FDCs, which is probably due to conjugating on average only one drug molecule to both the scFv fragment and the minibody monomer chain.

Antigen-binding properties of the FDCs were analyzed in direct ELISA. The generated FDCs based on both the scFv fragment and the minibody maintained binding to GD2 that was not statistically different from the parent antibody fragments (Figure 3F,G). These data demonstrate that conjugation with MMAE and MMAF does not affect the antigen-binding properties of the antibody fragments, and that the applied thiol-maleimide conjugation method is appropriate for generation of such FDCs.

### 2.3. Cytotoxic and Cytostatic Effects of the FDCs In Vitro

The B78-D14 melanoma cell line, which overexpresses GD2, was derived from the GD2-negative mouse melanoma cell line B16 by transducing genes responsible for GD2 biosynthesis [17]. Expression of GD2 on these cell lines was analyzed in our recent work [7]. Therefore, this pair of cell lines, differing mainly in GD2 expression, was used to evaluate their viability after treatment with the obtained FDCs in the MTT assay (Figure 4A,B). FDCs induced significant inhibitory effects in the B78-D14 cell line, in contrast to the GD2-negative B16 cell line. Moreover, FDCs based on both scFv fragments and minibodies conjugated with MMAF were more effective than those conjugated with MMAE (Figure 4A,B). For B78-B14 cells, the half maximal inhibitory concentrations (IC50) of scFv-MMAF and minibody-MMAF were 18.2 ± 4 nM and 9.7 ± 2 nM, respectively, whereas the IC50 of scFv-MMAE and minibody-MMAE constituted 187.2 ± 11 nM and 62.4 ± 6 nM, respectively, in the same cell line. Thus, regardless of the conjugated drug (MMAE or MMAF), the reduction in viability induced by FDCs based on minibodies was at least two times higher than that of the FDCs based on scFv fragments. It should be noted that the obtained FDCs had minimal effects, not even reaching 20% inhibitory concentration, in the B16 cell line that does not express GD2, which emphasizes the selectivity of the FDCs (Figure 4A,B).

GD2 is a recognized marker of neuroblastoma [18], so two human neuroblastoma cell lines, namely, GD2-negative NGP-127 and GD2-positive IMR-32, have been used to demonstrate FDC antitumor effects in this type of cancer. As we have previously shown, IMR-32 cells express GD2 at a high level, but significantly lower than B78-D14 cells, while NGP-127 cells do not express the marker [7,19]. The inhibitory effects of FDCs in the IMR-32 cell line were also significant in the MTT assay. However, FDCs based both on scFv fragments and minibodies were more effective in eliminating IMR-32 cells when they contained MMAE, in contrast to FDCs with MMAF which demonstrated a more prominent activity in B78-D14 cells (Figure 4C,D). For IMR-32 cells, the IC50 of scFv-MMAE and minibody-MMAE were 81.1 ± 5 nM and 56.5 ± 4 nM, respectively, and the IC50 of scFv-MMAF and minibody-MMAF were 116.7 ± 8 nM and 98.3 ± 6 nM, respectively. As with the case of GD2-negative mouse melanoma B16, cells of the GD2-negative human neuroblastoma NGP-127 were insensitive to the obtained FDCs even at high concentrations (Figure 4C,D). At the same time, free drugs did not demonstrate selective activity against GD2-positive and GD2-negative tumor cell lines; the IC50 of MMAE for all cell lines was roughly the same, approx. 1–1.5 nM (Figure 4E). Human neuroblastoma cell lines appeared to be somewhat more sensitive to MMAF (IC50 range 35–38 nM) compared to mouse melanoma lines (IC50 range 50–63 nM); however, the inhibitory response also did not correlate with GD2 levels (Figure 4F).

The MTT assay allowed us to assess the overall effect of drugs on cell viability without differentiating the mechanisms of action. In order to perform the latter, in addition to the MTT assay, we used the PI assay, which enabled us to study the contribution of cytotoxic and cytostatic effects to the overall influence of anti-GD2 FDCs on the tumor cells.

After treatment with FDCs, the number of cells with fragmented DNA in total populations of GD2-positive and GD2-negative cell lines was evaluated using the PI assay in order to confirm the inhibitory effects of the conjugates obtained in the MTT assay. For this, the same GD2-positive (B78-D14 and IMR-32) and GD2-negative (B16 and NGP-127) cell lines were incubated for 48 h with a constant concentration of scFv fragment-based FDCs (scFv-MMAE and scFv -MMAF) and minibody-based FDCs (Mb-MMAE and Mb-MMAF) (Figure 5A).

In the PI assay, all FDCs induced cell death in GD2-positive B78-D14 and IMR-32 cells, while none of them increased the level of cells with fragmented DNA in GD2-negative B16 and NGP-127 cell lines under the selected FDC concentration (100 nM) (Figure 5A). Minibody and scFv conjugates with MMAF showed the strongest effects in B78-D14 cells; the percentage of cells under the hypodiploid peak increased by 13 ± 3 and 9 ± 2 times, respectively, relative to the control. FDCs containing MMAE showed a less strong yet pronounced effect, with the percentage of cells with fragmented DNA treated with Mb-MMAE and scFv-MMAE increasing by 6 ± 1.5 and 4.5 times, respectively. The FDCs also demonstrated a pronounced increase in cell death in a different GD2-positive cell line, IMR-32, characterized by a lower level of GD2 expression; however, the effects were expectedly weaker than in the B78-D14 cell line overexpressing ganglioside GD2. For the IMR-32 line with a higher level of spontaneous apoptosis compared to other cell lines, the increase in cell death after treatment with the FDCs constituted 4.1x ± 0.6 for scFv-MMAE, 1.7x ± 0.9 for scFv-MMAF, 6.2x ± 1.2 for minibody-MMAE, and 2.2x ± 0.8 for minibody-MMAF, relative to the corresponding controls. The results from the PI assay indicate a greater efficiency of the conjugates with MMAF for the B78-D14 cell line, and of those with MMAE for the IMR-32 cell line, which is in agreement with the results obtained in the MTT assay. The results also confirm the absence of activity towards the GD2-negative B16 and NGP-127 lines at selected concentrations (Figure 4 and Figure 5A).

The MMAE and MMAF drugs used to generate the FDCs belong to the family of dolastins, the mechanism of activity of which comprises inhibition of microtubule polymerization that leads to arrest in the G2 phase of the cell cycle and subsequent cell death [20,21]. In order to confirm the effect of the FDCs on the cell cycle, we analyzed GD2-positive IMR-32 cells treated with the conjugates in the PI assay (Figure 5B). Figure 5B shows the distribution of cells with unfragmented DNA in the control samples and in the samples incubated with FDCs containing MMAE. It can be clearly seen that most of the control intact cells are at the G0/G1 stage of the cell cycle, while the FDCs induce the transition of cells with unfragmented DNA that have not yet entered the late stages of apoptosis to the G2/M stage of the cell cycle. Moreover, the minibody-MMAE which has the strongest effect on the viability of these cells causes the highest cell accumulation in the G2/M stage (Figure 5B). The data from the PI assay indicate that the mechanism of activity of the FDCs is directly related to the arrest of cells in the G2 phase of the cell cycle, and that the total inhibitory effect of the conjugates observed in the MTT assay consists of both the cytotoxic and cytostatic effects of MMAE or MMAF.

## 3. Discussion

Changing the format of the vector molecule in the ADCs from full-length antibodies to antibody fragments is an important strategy used to improve their efficacy [20,22]. Various antibody fragment formats can be employed to develop drug conjugates, including single domain antibody fragments (sdAb or VHH) [23], single chain variable fragments (scFv) [24], antigen-binding fragments (Fab) [25], or small immunoproteins (SIP) [26]. Non-IgG small scaffold proteins, such as affibodies, DARPins, or abdurins, can also serve as vector molecules within the conjugates [27,28].

Researchers are primarily guided by several criteria when choosing a small vector molecule format for developing FDCs. These include a balance between the circulation time of the vector molecule in the blood and its ability to penetrate into solid tumors. In addition, in the case of antibody fragments, it is necessary that they retain high affinity for the antigen and a significant rate of internalization, criteria which are also crucial for full-length antibodies [29]. An additional important factor for developing effective FDCs is the possibility of attaching a significant number of drug molecules to the protein molecule while maintaining the original antigen-binding properties [21,26].

It is often difficult to predict in advance which antibody fragment format will be optimal in a particular case, even for tumor markers of protein nature. In the context of the ganglioside GD2, a relatively small glycosphingolipid molecule located only on the outer leaflet of the plasma membrane, this aspect demands even more consideration. In addition, the number of research papers and experimental data regarding anti-GD2 ADCs is minimal. Therefore, we formulated two main goals in the present work: 1) to analyze the fundamental possibility of GD2-directed FDCs to induce cytotoxic antigen-specific effects in tumor cells, and 2) given that the aforementioned effects are present, to compare the efficacy of FDCs created on the basis of different variants of GD2-binding antibody fragments. Results from our previous studies served as prerequisites for the following work.

In our earlier paper, an scFv antibody fragment was generated based on the sequence of the ganglioside GD2-targeting murine antibody 14.18 which retained specific binding of the original antibody to GD2 [19]. We have also demonstrated that conjugates of the full-length chimeric antibody variant of 14.18 with the auristatin drugs MMAE and MMAF effectively eliminated GD2-positive tumor cells in vitro and strongly inhibited growth of GD2-expressing tumors in mice [7]. Since their smaller size compared to full-length antibodies typically leads to superior penetration of antibody fragments into solid tumors—a key factor influencing therapeutic efficacy [27]—we now set out to generate GD2-specific antibody fragment-drug conjugates based on the scFv fragment that we characterized earlier and to evaluate its ability to induce cell death in GD2-positive and GD2-negative tumor cell lines. As a second antibody fragment format besides scFvs that we employed for FDC generation, minibodies were chosen for possessing advantages such as bivalent binding with the antigen and increased blood circulation time relative to scFv fragments, while still having a smaller size compared to full-length antibodies [30].

FDCs based on both GD2-binding moieties, minibodies and scFv fragments, were successfully generated by site-direct conjugation to MMAE or MMAF via a cathepsin-B cleavable maleimide linker and with specific drug-to-antibody ratios (DAR 2 for minibody-based FDCs; DAR 1 for scFv-based FDCs). We confirmed specific binding of the FDCs to ganglioside GD2 in direct ELISA and to GD2-positive cells in flow cytometry and confocal microscopy analyses. Both formats induced selective cytotoxic and cytostatic effects in GD2-positive but not GD2-negative neuroblastoma and melanoma cell lines. The IC50 values for both formats were in the nanomolar range in GD2-positive tumor cell lines.

When analyzed in the MTT assay, the FDCs induced significant inhibitory effects in the B78-D14 mouse melanoma cell line that overexpresses GD2, in contrast to the GD2-negative B16 mouse melanoma cell line. Moreover, the FDCs based on both scFv fragments and minibodies conjugated to MMAF manifested stronger cytotoxic effects in this cell line than those conjugated to MMAE. The inhibitory effects of the FDCs were also significant in the GD2-positive human neuroblastoma cell line IMR-32. However, the FDCs based both on scFv fragments and minibodies were more effective in eliminating IMR-32 cells when they contained MMAE, in contrast to the FDCs with MMAF, which demonstrated a more prominent activity in B78-D14 cells. As in the case of GD2-negative mouse melanoma B16, cells of the GD2-negative human neuroblastoma NGP-127 were insensitive to our fragment-drug conjugates even at high concentrations.

In order to differentiate the mechanisms of action of the FDCs, we used the PI assay which enabled us to study the contribution of cytotoxic and cytostatic effects to the overall activity of the anti-GD2 FDCs on the tumor cells. All FDCs induced cell death in GD2-positive B78-D14 and IMR-32 cells, while none of them increased the level of cells with fragmented DNA in GD2-negative B16 and NGP-127 cell lines. Minibody and scFv conjugates with MMAF showed the strongest effects in B78-D14 cells; FDCs containing MMAE manifested a less strong yet a pronounced effect. The FDCs also demonstrated a marked increase in cell death in a different GD2-positive cell line, IMR-32, characterized by a lower level of GD2 expression; however, the effects were expectedly weaker than in the B78-D14 cell line overexpressing ganglioside GD2. The results from the PI assay indicated a greater efficiency of the conjugates with MMAF for the B78-D14 cell line, and of those with MMAE for the IMR-32 cell line, which is in agreement with the results obtained in the MTT assay. The results also confirmed the absence of activity towards the GD2-negative B16 and NGP-127 lines at selected concentrations.

We next set out to evaluate the effects of our FDCs on the cell cycle, for which the GD2-positive IMR-32 cells treated with the conjugates were analyzed in the PI assay. The distribution of cells with unfragmented DNA in the control samples and in the samples incubated with the FDCs containing MMAE clearly showed that most control intact cells were at the G0/G1 stage of the cell cycle, while the FDCs induced the transition of cells with unfragmented DNA that had not yet entered the late stages of apoptosis and had unfragmented DNA to the G2/M stage of the cell cycle. Moreover, the minibody-MMAE which had the strongest effect on the viability of these cells caused the highest accumulation in the G2/M stage. The data from the PI assay indicate that the mechanism of activity of the FDCs is directly related to the arrest of cells in the G2 phase of the cell cycle and that the total inhibitory effect of the conjugates observed in the MTT assay consists of both the cytotoxic and the cytostatic effects of MMAE or MMAF.

The minibody-based FDCs with both MMAE and MMAF demonstrated more pronounced cytotoxic effects compared to scFv-based FDCs in all GD2-positive tumor cell lines within the analysis. This can be explained both by the higher DAR in minibody-based FDCs and by their stronger antigen-binding properties, resulting from the bivalent interaction of the minibodies with the antigen in contrast to monovalent interaction for scFv fragments. The valency of the interaction is an important criterion for the high binding affinity of an antibody or antibody fragment when interacting with ganglioside GD2, a carbohydrate antigen [31]. In this regard, approaches that enhance the cytotoxic activity of FDCs by increasing the valency of binding to GD2, for example, by multimerization of the antigen-binding moieties [29], or by increasing the affinity of binding to GD2 by affinity maturation [32], are of particular interest. The FDCs obtained in this work are inferior in efficacy to the anti-GD2 full-length antibody-drug conjugates studied by us before [7]. One reason for the reduced efficiency of the FDCs may be found in their lower DAR compared to the ADCs (specifically, DAR 2 for the minibody-based FDCs versus DAR 4 for the ADCs). Design of methodologies for increasing DAR, optimization of antigen-binding moieties that facilitate the affinity of interaction with GD2 and internalization of the FDC-GD2 complex into the cell, and innovative linkers and cytotoxic drugs are all among the strategies that could contribute to developing GD2-specific antibody fragment-drug conjugates applicable for clinical translation.

## 4. Materials and Methods

### 4.1. Expression and Purification of scFv Fragments and Minibodies

ScFv fragments of GD2-specific antibodies 14.18 were produced by transient gene expression in modified HEK293 cells (Expi293 Expression system, Thermo Fisher Scientific, Waltham, MA, USA), as described previously [18].

Minibodies were produced in the same expression system. To this end, the minibody monomer sequence was cloned between the XbaI and AgeI restriction sites of the eukaryotic expression vector pcDNA3.4, followed by verification of the construct identity by analytical restriction digests. The vector was then purified by QIAGEN Plasmid Maxi Kit (Qiagen, Hilden, Germany), and transfection was performed in 50 mL cell culture volume following the protocol for the expression system. The cell culture was routinely checked for optical density and the percentage of viable cells, while the target protein content in the medium was assessed by polyacrylamide gel electrophoresis. The Antibody-Expressing Positive Control Vector from the expression system kit was employed for additional monitoring of protein expression level under recommended conditions.

ScFv fragments and minibodies were isolated from the cell culture supernatants by protein L chromatography (HiTrap Protein L purification column, GE Healthcare, Chicago, IL, USA) as described previously [18], followed by size-exclusion chromatographic purification in phosphate-buffered saline (PBS) eluent at 0.5 mL/min flow rate on a Superdex 200 10/300 GL column (GE Healthcare, Chicago, IL, USA), with detection at 280 nm by Beckman System Gold HPLC system. The fragments were concentrated on Amicon Ultra 10 kDa filters (Merck, Rahway, NJ, USA) and sterilized through 0.22 μm membrane filters. Protein concentration was calculated at a wavelength of 280 nm by BioDrop μLITE spectrophotometer (BioChrom, Cambridge, UK).

### 4.2. FDC Generation

Conjugation of the scFv fragments and the minibodies to MC-VC-PABC-MMAE or MC-VC-PABC-MMAF (both from BOC Sciences, Shirley, NY, USA), further referred to as MMAE or MMAF, was carried out by thiol-maleimide chemistry. First, reduction of the C-terminal cysteine introduced intro the scFv fragment and the cysteines forming disulfide bonds between minibody chains was performed by 0.5 mM Tris(2-carboxyethyl)phosphine (TCEP) in PBS buffer, for 1.5 h at room temperature (RT) and with agitation. The reducing agent was then removed by a Zeba Spin Desalting Column, 7K MWCO (Thermo Fisher Scientific, Waltham, MA, USA), followed by an immediate reaction of 1–2 mg/mL proteins with 4:1 molar excess of the corresponding auristatin drug over each antibody fragment (for FDCs with optimized DAR) in PBS buffer at pH 6.0 with addition of 0.2 g/l EDTA, for 3 h at RT and with agitation. The percent of DMSO from the drug stock solutions was kept below 3% in the reaction mix. The FDCs were then transferred to PBS in Zeba columns. Both scFv fragments and minibodies were additionally conjugated to fluorescein 6-maleimide (Lumiprobe, Moscow, Russia), further referred to as FAM, in reaction conditions identical to those described above.

Product purity and stability were analyzed by size-exclusion chromatography, as described above, and by gel electrophoresis. Visualization of the naked antibody fragments and the FDCs by SDS-PAGE was performed either in reducing or in non-reducing conditions, with modifications to [33]. For reducing electrophoresis, proteins in standard SDS sample buffer containing 50 mM dithiothreitol were heated to 95 °C for either 5 or 10 min and loaded onto the gels. For non-reducing electrophoresis, no dithiothreitol was added to the sample buffer and no heating was performed, but the samples in the buffer were vortexed and incubated for 15 min prior to loading the gel. Samples were resolved in 10% gels (NuPAGE Mini Protein Gels, Thermo Fisher Scientific, Waltham, MA, USA), and the gels were stained with Coomassie R250 and analyzed in Gel Doc EZ Imager and Image Lab software (Bio-Rad, Hercules, CA, USA).

### 4.3. Evaluation of Drug-to-Antibody Ratio

Average drug-to-antibody ratio (DAR) for antibody fragments conjugated with MMAE or MMAF, as well as degree of labeling (DOL) for conjugates with FAM-maleimide, were calculated by UV-VIS spectroscopy (Table 1) on a BioDrop µLITE spectrophotometer, as described by Chen [34]. The absorbance values at 253 nm wavelength, which was the absorption maximum for MC-VC-PABC-MMAE and MC-VC-PABC-MMAF, and at 280 nm were used for calculating the DAR of the auristatin-carrying FDCs. Extinction coefficients employed in the analysis are presented in the table below.

### 4.4. Direct ELISA

Nunc MaxiSorp high protein-binding capacity 96-well ELISA plates (Thermo Fisher Scientific, Waltham, MA, USA) were coated with gangliosides GD2, GM2, GD1b, and GD3 at a concentration of 0.25 μg in 100 μL of 96% ethanol per well. GD2, GD1b, and GD3 were purchased from Sigma-Aldrich, and GM2 was obtained according to our previous work [35]. Following air drying, plate wells were blocked with 100 μL 2% BSA in PBS supplemented with 0.1% Tween-20 (PBS-T) per well for 2 h at RT. ScFv fragments, minibodies, or FDCs (100 μL solution in PBS-T per well) were added in triplicates at different concentrations, and incubation was carried out for 1.5 h. After washing with PBS-T, anti-FLAG HRP-labeled antibodies were added to the wells with scFv fragments, and anti-human Fc-specific HRP-labeled antibodies were added to the wells with minibodies (both 1:6000; Santa Cruz Biotechnology, CA, USA). After 40 min incubation and further washing, 1-Step Ultra TMB-ELISA Substrate Solution (Thermo Fisher Scientific, Waltham, MA, USA) was added, and the color reaction OD was measured at 450 nm by Multiscan FC microplate reader (Thermo Fisher Scientific, Waltham, MA, USA). Percent of cross-reactivity was calculated as the ratio of TMB color reaction OD_450_ in GM2-, GD1b-, or GD3-coated wells to OD_450_ in GD2-coated wells.

### 4.5. Cell Lines

B16 and B78-D14 mouse melanoma cell lines were cultured in RPMI-1640, IMR-32 human neuroblastoma in EMEM, and NGP-127 human neuroblastoma in DMEM medium. All culture media were supplemented with 10% heat-inactivated fetal bovine serum, 2 mM *L*-glutamine, 100 μg/mL penicillin, and 100 U/mL of streptomycin (all–Gibco, Waltham, MA, USA).

IMR-32 and B16 cell lines were obtained from ATCC, and the NGP-127 cell line was kindly provided by Anton Buzdin (Shemyakin–Ovchinnikov Institute of Bioorganic Chemistry RAS, Moscow, Russia). The GD2-positive B78-D14 mouse melanoma cell line generated by transfection of the GD2-negative B16 line with genes coding for GD3 and GD2 synthases [17] was a kind gift from David Schrama (University Hospital Wuerzburg, Wuerzburg, Germany). All cell lines were maintained at low passage numbers and routinely checked for Mycoplasma by PCR. Human cell lines were authenticated by STR analysis.

### 4.6. Flow Cytometry

Staining of B16, B78-D14, IMR-32, and NGP-127 cells with FAM-labeled scFv fragments and minibodies was performed as described previously [36]. In brief, cells were detached from the culture plates (cells were trypsinized and washed twice in PBS), incubated with FAM-labeled antibody fragments (1 µg per sample) for 1 h in PBS supplemented with 1% FBS and 0.02% sodium azide, and then washed twice in PBS. All procedures were performed at 4°C. The samples were immediately analyzed using BD FACSCalibur flow cytometer (Becton Dickinson, USA). In each sample, at least 10,000 events were collected. For all samples, the analysis was performed in triplicate. The relative fluorescence intensity (RFI) of GD2 expression in cell lines was calculated as the ratio of specific fluorescence of cells stained with FAM-labeled antibody fragments to autofluorescence of control unstained cells. The data were analyzed using FlowJo and WinMDI software.

### 4.7. Confocal Microscopy

Staining of GD2-positive and GD2-negative cell lines for immunofluorescence analysis was performed by a modified protocol from our previous work [37]. In brief, 200,000 B16 or B78-D14 cells were seeded on glass coverslips (Fisher Scientific, Waltham, MA, USA) and grown to 80% confluence in 6-well tissue culture plates (Greiner, Kremsmünster, Austria). The cells were then washed with PBS and incubated with the FAM-labeled scFv fragment, minibody, or full-length ch14.18 antibody (1 μg per sample) with addition of Hoechst 33342 nuclear counterstain (0.5 μg per sample) for 1 h at 4°C in PBS supplemented with 1% FBS and 0.02% sodium azide. The FAM-labeled ch14.18 antibody was obtained according to our previous work [7], essentially by the same method as the FAM-labeled antibody fragments generated here. Following a wash in PBS, the cells were fixed with 4% paraformaldehyde for 30 min at RT, thoroughly rinsed in PBS, and mounted in Mowiol 4–88 (Merck, Rahway, NJ, USA). Slides were analyzed using an EZ-C1 Eclipse TE2000 confocal laser scanning microscope (Nikon, Tokyo, Japan) equipped with Plan Apo 40X and 60X objectives. Images were collected with EZ-C1 program and processed with EC1 Viewer (Nikon, Tokyo, Japan).

### 4.8. MTT Assay

FDC-induced decrease in cell viability was analyzed by colorimetric MTT (3-[[4,5]-dimethylthiazol-2-yl]-2,5-diphenyltetrazolium bromide; purchased from Sigma-Aldrich, St. Louis, MO, USA) assay previously described by Denizot and Lang [38], with modifications specified earlier [39]. Briefly, tumor cells were cultured in 96-well flat-bottom tissue culture plates (2 × 10^4^ cells/well, Greiner, Kremsmünster, Austria) with serial dilutions of FDCs for 72 h under standard culture conditions. Following incubation, the MTT solution (final concentration 250 μg/mL) was added to each sample for 3 h. Reaction OD was assessed by Multiscan FC microplate reader at a wavelength of 540 nm. Cell viability was calculated using the formula (OD_treated cells_ − OD_blank_)/(OD_control cells_ − OD_blank_) × 100%, where OD_blank_ represents OD in control wells containing no cells. SigmaPlot (Systat Software Inc., San Jose, CA, USA) was used to generate dose–response curves. All MTT experiments were reproduced at least three times.

### 4.9. Propidium Iodide Assay for Cell Death and Cell Cycle Analysis

Analysis of cell death or cell cycle was performed using propidium iodide (PI) staining in accordance with modifications [40] to the previously described method [41]. Tumor cells (10^6^ cells per sample) were incubated with FDCs at concentration 100 nm for 48 h under standard culture conditions. Cells were subsequently fixed and permeabilized with ice-cold ethanol for 60 min at 4°C, and washed twice with PBS by centrifugation for 10 min at 300× *g*. Cell pellets were resuspended in DNA staining buffer (PBS supplemented with 20 μg/mL PI (Sigma-Aldrich, St. Louis, MO, USA), 20 μg/mL RNase A (Thermo Fisher Scientific, Waltham, MA, USA)) and further incubated for 30 min at 4 °C. For all samples, cell death and cell cycle were performed in triplicate. In each sample, at least 10,000 events were collected. The quantitation of DNA content was evaluated with BD FACSCalibur flow cytometer (Becton Dickinson, Franklin Lakes, NJ, USA).

### 4.10. Statistical Analysis

Graphs were created using SigmaPlot, GraphPad Prism, and MS Excel software. The data are represented as mean ± SEM of at least three independent experiments, or as one representative experiment from three. Statistical analysis was performed by unpaired Student’s t-test. Significance levels of *p* < 0.05 were considered statistically reliable.

## Figures and Tables

**Figure 1 ijms-24-01239-f001:**
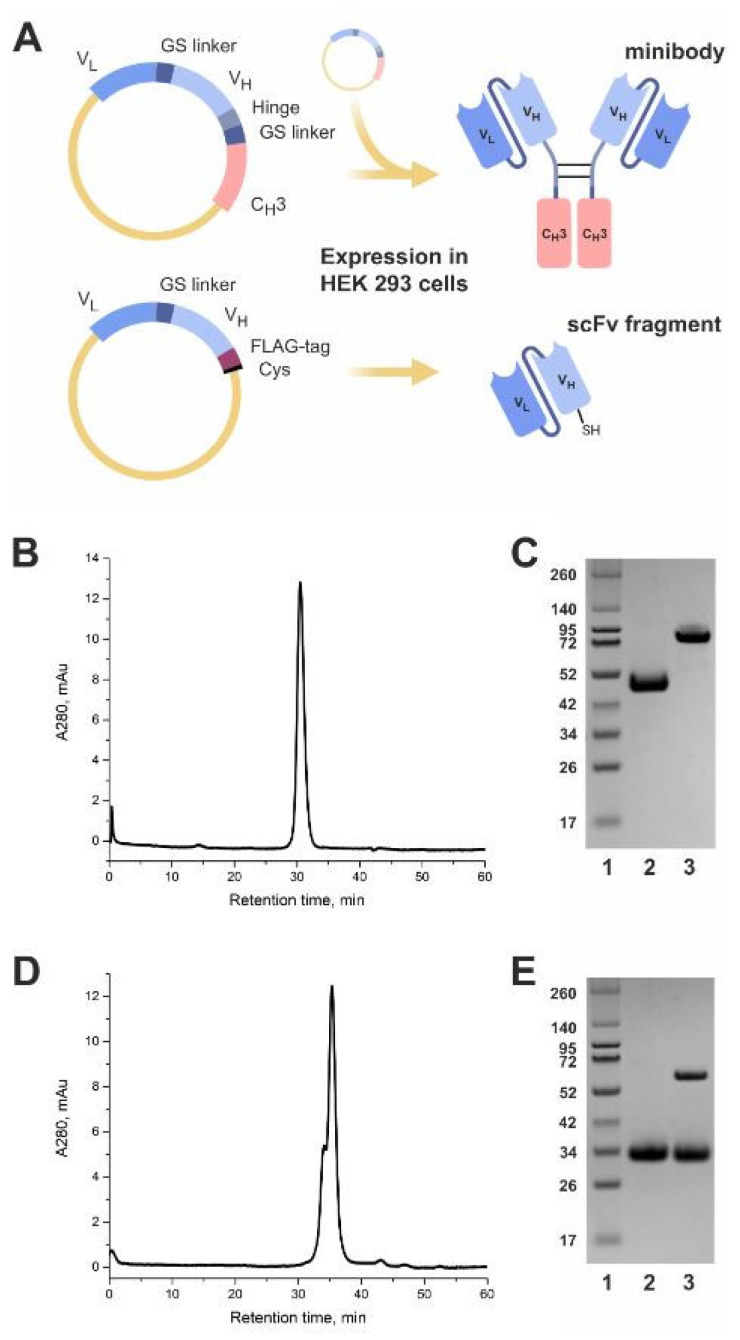
Design and expression of the GD2-specific antibody fragments. (**A**) Structures of the minibody 14.18 and the scFv fragment 14.18. (**B**) Size-exclusion chromatographic analysis of the minibody. (**C**) Polyacrylamide gel electrophoresis of the minibody; 1, molecular weight protein markers; 2, the minibody under reducing conditions, 10 min incubation at 95 °C; 3, the minibody under non-reducing conditions. (**D**) Size-exclusion chromatographic analysis of the scFv. (**E**) Polyacrylamide gel electrophoresis of the scFv; 1, molecular weight protein markers; 2, the scFv under reducing conditions, 10 min incubation at 95 °C; 3, the scFv under non-reducing conditions.

**Figure 2 ijms-24-01239-f002:**
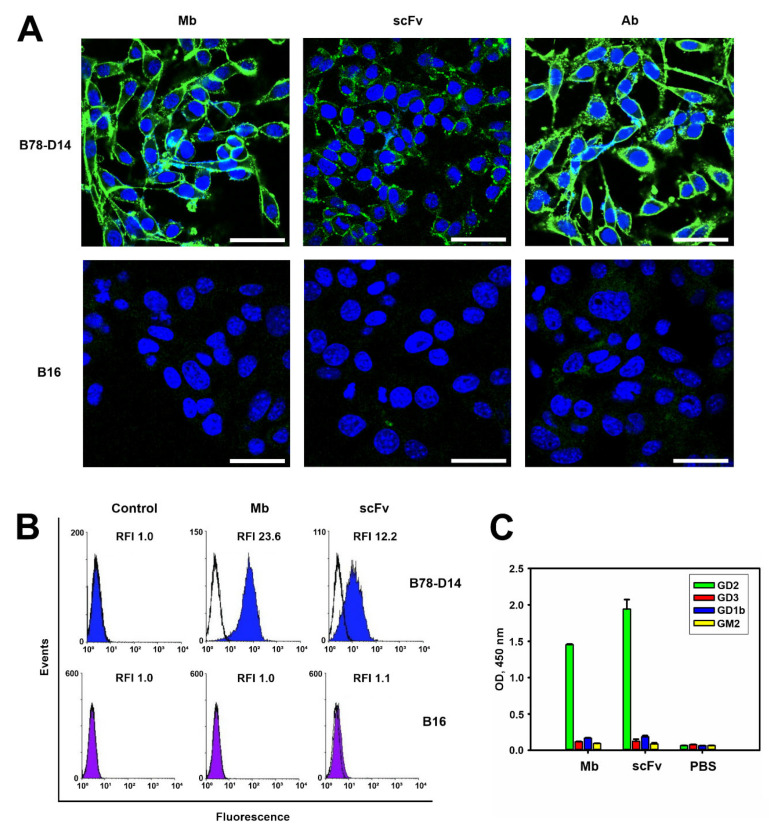
Antigen-binding properties of antibody fragments. (**A**) Confocal images of GD2-positive B78-D14 and GD2-negative B16 melanoma cell lines stained with FAM-labeled minibodies, scFv fragments, or control ch14.18 antibodies (1 µg per sample). Staining with fluorescently labeled anti-GD2 antibodies or antibody fragments (green) and counterstaining the nuclei with Hoechst 33342 (blue). Bar scale: 10 μm. (**B**) Flow cytometry analysis. Staining of B78-D14 and B16 cell lines with FAM-labeled minibodies and scFv fragments (1 µg per sample). Control–fluorescence intensity of intact cells. (**C**) Evaluation of the binding of minibodies and scFv fragments to gangliosides GD2, GD3, GM2, and GD1b in direct ELISA. Gangliosides were adsorbed on the plate; antibody fragments were added to the wells in 5 nM concentration. After incubating scFv fragments with anti-FLAG antibodies and minibodies with anti-human Fc-specific antibodies (both at 1:6000), the reaction was developed by Ultra TMB-ELISA Substrate Solution. Data are represented as OD mean ± SEM. scFv–scFv fragments 14.18, Mb–minibodies 14.18, Ab–antibodies 14.18.

**Figure 3 ijms-24-01239-f003:**
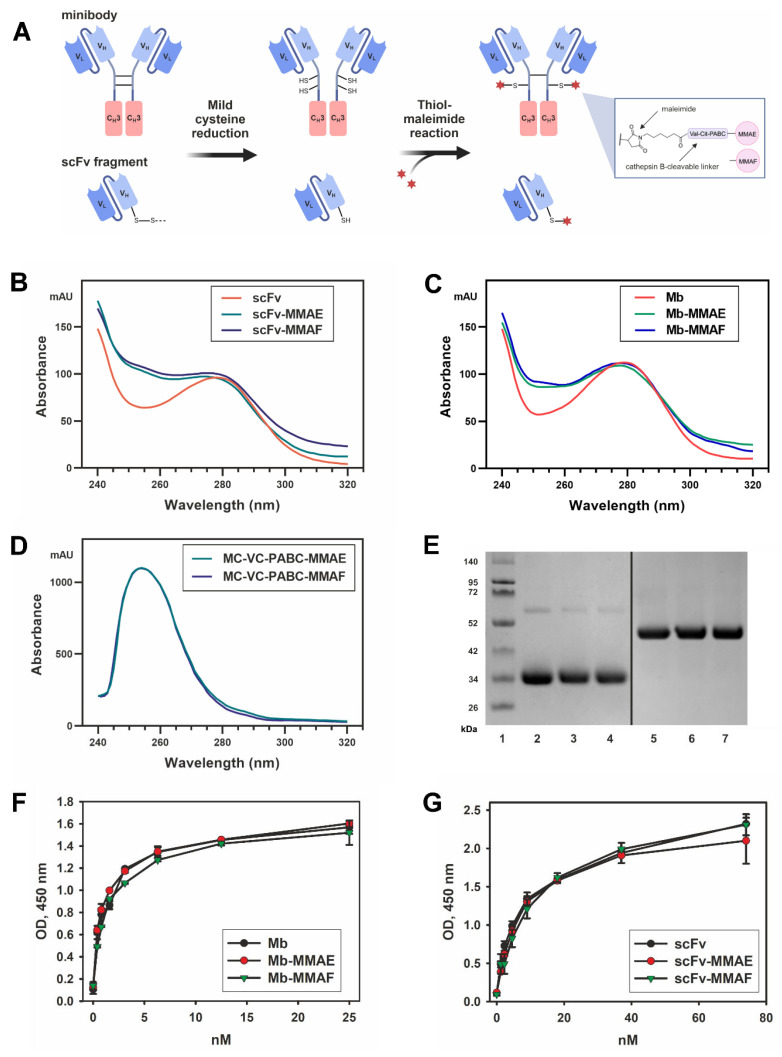
Production of the FDCs and analysis of their antigen-binding properties. (**A**) Reaction scheme of the generation of scFv and minibody conjugates with MMAE and MMAF. (**B**) Absorption spectra of the scFv-based FDCs normalized at 280 nm. (**C**) Absorption spectra of the minibody-based FDCs normalized at 280 nm. (**D**) Absorption spectra of the auristatin drugs used for conjugation. (**E**) Polyacrylamide gel electrophoresis of the FDCs in reducing conditions, 5 min incubation at 95 °C. 1, molecular weight protein markers; 2, scFv; 3, scFv-MMAE; 4, scFv-MMAF; 5, minibody; 6, minibody-MMAE; 7, minibody-MMAF. (**F**) Direct ELISA of minibodies and minibody-based FDCs; ganglioside GD2 was adsorbed on the plate, serial dilutions of scFv fragments were added to the wells; after incubation with HRP-labeled anti-human antibodies (1:6000), the reaction was developed by Ultra TMB-ELISA Substrate Solution. Data are represented as mean ± SEM. (**G**) Direct ELISA of scFv fragments and scFv-based FDCs; ganglioside GD2 was adsorbed on the plate, serial dilutions of scFv fragments were added to the wells; after incubation with HRP-labeled anti-FLAG antibodies (1:6000), the reaction was developed by Ultra TMB-ELISA Substrate Solution. Data are represented as mean ± SEM. scFv–scFv fragments 14.18, Mb–minibodies 14.18.

**Figure 4 ijms-24-01239-f004:**
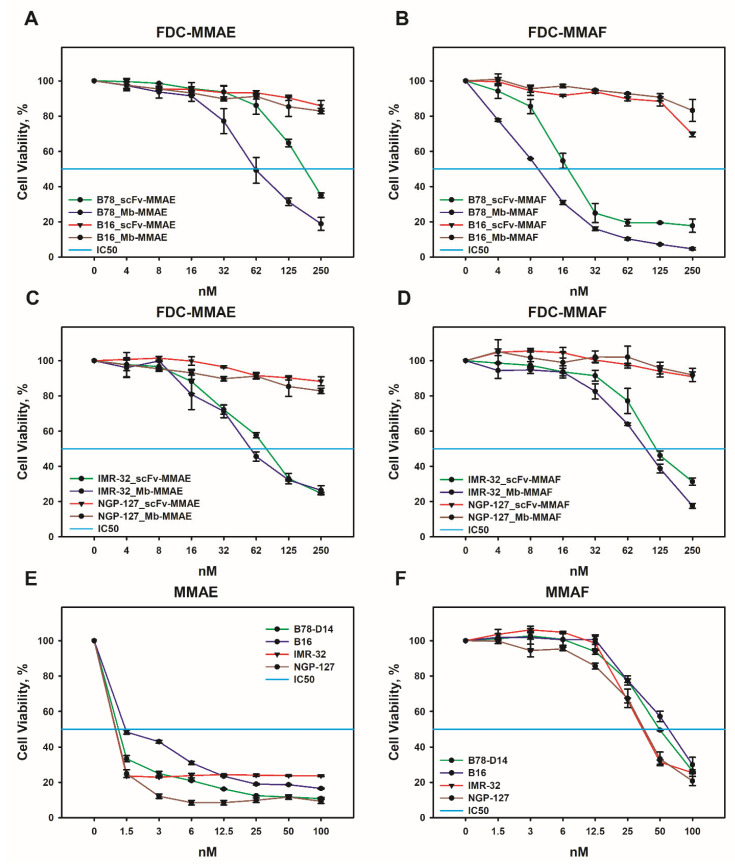
Cytotoxic activity of anti-GD2 FDCs in GD2-positive (B78-D14 and IMR-32) and GD2-negative (B16 and NGP-127) cell lines. Viability of cell lines analyzed by MTT assay following 72 h incubation with scFv-MMAE and minibody-MMAE (**A**,**C**)**,** scFv-MMAF and minibody-MMAF (**B**,**D**), MMAE (**E**), or MMAF (**F**). scFv–scFv fragments 14.18, Mb–minibodies 14.18.

**Figure 5 ijms-24-01239-f005:**
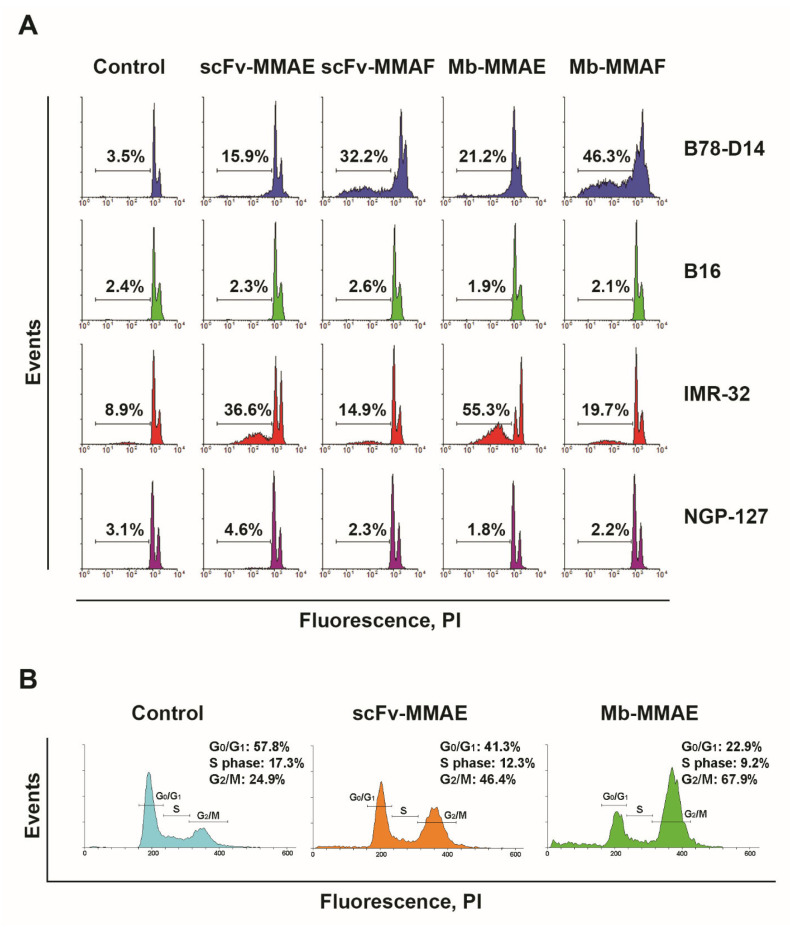
PI assay. (**A**) Analysis of DNA fragmentation (propidium iodide staining) of GD2-positive (B78-D14 and IMR-32) and GD2-negative (B16 and NGP-127) cell lines treated with FDCs (100 nM) for 48 h. Control—untreated control cells. The percentage of cells with fragmented DNA in hypodiploid peaks is shown for each histogram. (**B**) Cell cycle analysis (propidium iodide staining) of GD2-positive human neuroblastoma IMR-32 treated with FDC-MMAE (100 nM) for 48 h. Control—untreated control cells. scFv–scFv fragments 14.18, Mb–minibodies 14.18.

**Table 1 ijms-24-01239-t001:** Extinction coeffficients employed for calculation of the drug-to-antibody ratios.

Molecule	Extinction Coefficients (cm^−1^ M^−1^) Calculated at Given Wavelength		
	253 nm	280 nm	494 nm
scFv 14.18	26,460	39,960	8640
minibody 14.18	64,590	129,170	14,680
MC-VC-PABC-MMAE	21,920	3220	-
MC-VC-PABC-MMAF	21,900	2740	-
fluorescein 6-maleimide	-	12,580	74,000

## Data Availability

The data presented in this study are available within the article text and figures.

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
