# Peer review of "Minibody-Based and scFv-Based Antibody Fragment-Drug Conjugates Selectively Eliminate GD2-Positive Tumor Cells"

_ijms, 2023, doi:10.3390/ijms24021239_

Round 1

Reviewer 1 Report

You have submitted « Antibody fragment drug-conjugates for the treatment of GD2-positive tumors “ by Kalinovski and coworkers in IJMS. This is an interesting work highlighting the ADCs targeting GD2 in neuroectoderm derived tumors. Using the drugs monomethyl auristatin E and F, you showed a selective cytotoxicity in tumor cells lines and GD2 positive cancer models.

However, the title you propose remains too large and does not emphasize the type of ADCs you used and the result you obtained. The authors can reconsider the title to add more precision.

The authors did not provide the full name of the abbreviation you used in the text, for example, PI assays line 102. The authors should check all the abbreviation.

Furthermore, you did not provide a description/ explanation of ganglioside in the introduction part. For readers, gangliosides could be an unknown family of glycolipid. The authors should add a paragraph in the introduction to explain this molecule precisely.

The analysis of few cell lines is not conclusive. In particular, the authors used a pair of cells GD2 positive and negative cell lines without demonstrating the expression. The actually relevant data is missing concerning the expression of GD2 for example using antibodies.

In figure 1, the authors showed “HEK293” as cellular model while in the results line 124, authors used “Expi293F”. Which cell line did they use? Authors should be consistent in the term for designing cellular model.

In Figure 2, authors does not explained clearly what Mb is. Besides, the authors should explained how they choose the concentration of nanobody for their experiment. The same comment is available for Figure 5.

The discussion/conclusion part have to be more developed considering the results obtained. Authors should emphasizes all the results they obtained.

Author Response

Thank you very much for the detailed analysis of our manuscript and valuable suggestions. We have revised the manuscript accordingly and hope that its quality has improved. We also introduced a number of language corrections to the text.

Please, find our point-by-point response (AU) to the questions and comments below. These changes are also indicated by highlighted text in the revised manuscript files submitted for review.

Reviewer 1

You have submitted «Antibody fragment drug-conjugates for the treatment of GD2-positive tumors“ by Kalinovski and coworkers in IJMS. This is an interesting work highlighting the ADCs targeting GD2 in neuroectoderm derived tumors. Using the drugs monomethyl auristatin E and F, you showed a selective cytotoxicity in tumor cells lines and GD2 positive cancer models.

Response to Reviewer 1 Comments

Point 1: However, the title you propose remains too large and does not emphasize the type of ADCs you used and the result you obtained. The authors can reconsider the title to add more precision.

AU: We thank the Reviewer for pointing out the need to change the title. We have changed it to more accurately reflect the topic of the article, primarily specifying the antibody fragments used in the FDCs.

Point 2: The authors did not provide the full name of the abbreviation you used in the text, for example, PI assays line 102. The authors should check all the abbreviation.

AU: We re-checked the abbreviations in the text. Specifically, descriptions for the following abbreviations were added:

ELISA (Line 99); PI (Line 102); FLAG octapeptide tag (Line 110); TMB (Line 175); OD (Line 176); IC50 (Line 244); PBS (Line 415).

Point 3: Furthermore, you did not provide a description/ explanation of ganglioside in the introduction part. For readers, gangliosides could be an unknown family of glycolipid. The authors should add a paragraph in the introduction to explain this molecule precisely.

AU: We added a paragraph highlighting the key information about GD2 in the introduction, and provided a citation to an additional relevant review article on the topic.

Point 4: The analysis of few cell lines is not conclusive. In particular, the authors used a pair of cells GD2 positive and negative cell lines without demonstrating the expression. The actually relevant data is missing concerning the expression of GD2 for example using antibodies.?

AU: In this work, we used cell lines that we analyzed for GD2 expression in our previous publications (for example, in https://doi.org/10.3390/molecules24213835 and https://doi.org/10.1136/jitc-2022-004646). The data obtained by us are in good agreement with the data on GD2 expression on these lines from other authors [https://doi.org/10.1016/j.canlet.2009.02.040], and the B78-D14 cell line was specifically created as a GD2-positive line [doi: 10.1073/pnas.91.22.10455]. Following the reviewer’s recommendation, we included confocal staining data for the B16 and B78-D14 cell lines with the FAM-labeled full-length antibody dinutuximab (Figure 2A) and added references to our earlier work, in which we assessed GD2 expression on all cell lines used in this work.

Point 5: In figure 1, the authors showed “HEK293” as cellular model while in the results line 124, authors used “Expi293F”. Which cell line did they use? Authors should be consistent in the term for designing cellular model.

AU: We made an adjustment to the methods section to indicate that Expi293F is the commercial name used by Thermo Scientific for their optimized HEK293 line.

Point 6: In Figure 2, authors does not explained clearly what Mb is.

AU: We checked the abbreviations in the legend of Figure 2 and also introduced an additional abbreviation for the ch14.18 full-length antibody added to the confocal microscopy analysis.

Point 7: Besides, the authors should explained how they choose the concentration of nanobody for their experiment. The same comment is available for Figure 5.

AU:  In confocal and cytometry experiments, we routinely use the labeled antibodies in a concentration of 1 µg per sample (or 10 µg/mL for a sample volume of 100 µL). This concentration is usually sufficient to bind to all surface antigens for which antibodies are specific. We have standardized the way we indicate the antibody concentrations for these two methods in Figure 2. For ELISA, a representative experiment with one selected concentration is presented, while the absence of cross-reactivity was observed at all evaluated concentrations of both the scFv fragments and minibodies. We have included an additional clarification in the text of the article on this issue.

Several concentrations of FDCs were initially used in the PI assay. However, the high concentration of the FDCs (100 nM) presented in the manuscript made it possible to clearly identify and separate the cytotoxic effects (induction of cell death) and the cytostatic effects (cell cycle arrest) under the action of the different FDCs generated in our work. Therefore, we present the results with this particular FDC concentration.

Point 8: The discussion/conclusion part have to be more developed considering the results obtained. Authors should emphasizes all the results they obtained.

AU: We have introduced additional points to the discussion section of the work to better reflect the results according to the Reviewer’s suggestions.

Reviewer 2 Report

Kalinovsky et al have worked on investigating the efficacy of antibody fragment drug conjugates for GD2 positive cancer tumors. This analysis seems to be the first of its kind that looks at the FDCs for GD2 positive tumor therapy. The authors worked on quantifying the FDC binding as well as their cytotoxic capabilities for GD2 negative and positive cancer cell lines. Overall it is well written manuscript going on every data in much details.

There are a few opportunities to improve the clarity and impact of the manuscript.

Page 4 Line 150 - Cell lines used here GD2 positive and negative– western blots or flow cytometry showing the relative amount of GD2 in each of these cell lines would be helpful to manuscript. This analysis should also be done for the cell lines used in the next experiments so that the efficacy of FDCs can be corelated with the amount of GD2 present.

Page 8 Line 242-242 – These results look very promising and they seem to indicate that minibodies are better compared to scFvs. Wouldn’t it add more value to the manuscript to show a comparison of these FDCs with the parent antibody as an ADC?

In vivo studies of these FDCs would help in improving the scope and impact of the manuscript.

Author Response

Thank you very much for the detailed analysis of our manuscript and valuable suggestions. We have revised the manuscript accordingly and hope that its quality has improved. We also introduced a number of language corrections to the text.

Please, find our point-by-point response (AU) to the questions and comments below. These changes are also indicated by highlighted text in the revised manuscript files submitted for review.

Reviewer 2.

Kalinovsky et al have worked on investigating the efficacy of antibody fragment drug conjugates for GD2 positive cancer tumors. This analysis seems to be the first of its kind that looks at the FDCs for GD2 positive tumor therapy. The authors worked on quantifying the FDC binding as well as their cytotoxic capabilities for GD2 negative and positive cancer cell lines. Overall it is well written manuscript going on every data in much details.

There are a few opportunities to improve the clarity and impact of the manuscript.

Response to Reviewer 2 Comments

Point 1. Page 4 Line 150 - Cell lines used here GD2 positive and negative– western blots or flow cytometry showing the relative amount of GD2 in each of these cell lines would be helpful to manuscript. This analysis should also be done for the cell lines used in the next experiments so that the efficacy of FDCs can be corelated with the amount of GD2 present.

AU:

In this work, we used cell lines that we analyzed for GD2 expression in our previous publications (for example, in https://doi.org/10.3390/molecules24213835 and https://doi.org/10.1136/jitc-2022-004646). The data obtained by us are in good agreement with the data on GD2 expression on these lines from other authors [https://doi.org/10.1016/j.canlet.2009.02.040], and the B78-D14 cell line was specifically created as a GD2-positive line [doi: 10.1073/pnas.91.22.10455]. Following the reviewer’s recommendation, we included confocal staining data for the B16 and B78-D14 cell lines with the FAM-labeled full-length antibody dinutuximab (Figure 2A) and added references to our earlier work, in which we assessed GD2 expression on all cell lines used in this work.

Point 2. Need to define that B78-D14 is a melanoma cell line as it isn't included in the section where the other cell lines are defined.

AU: We have additionally emphasized in the results section that B78-D14 cell line is a melanoma (Line 257).

Point 3. Page 8 Line 242-242 – These results look very promising and they seem to indicate that minibodies are better compared to scFvs. Wouldn’t it add more value to the manuscript to show a comparison of these FDCs with the parent antibody as an ADC?

AU: Thank you for a very justified question. The goal of this work was to conceptually demonstrate that antibody fragments can be used within drug conjugates to eliminate GD2-positive tumor cells, since such research has not been conducted before. ADCs based on the parental full-length antibody dinutuximab have manifested significantly stronger antitumor effects in our experiments so far (please see 10.1136/jitc-2022-004646). We deliberately do not give a detailed comparison of the ADCs and the FDCs, although we mention the more pronounced activity of the ADCs compared to the FDCs generated in this work in the Discussion section. We see several reasons for such a difference. First, the developed FDCs have a smaller DAR compared to the ADCs from our earlier work (both contain the same MMAE/MMAF drugs conjugated via an identical cleavable linker). According to our data, if the conjugates contain 1 or 2 drug molecules (MMAE used in our experiments), the difference in the observed in vitro effects for the ADCs and FDCs is insignificant. And at the same time, the effects of the FDCs with DAR 2 (the maximum DAR for the minibody-based FDC that does not compromise its stability, for conjugation via interchain cysteines) vary significantly from the effects of ADCs with DAR 4-6 (typical DAR for conjugates of IgG molecules with auristatin drugs). We are working on strategies to conjugate more drugs to antibody fragments and/or employ more cytotoxic drugs (e.g. PBD – we have already ordered these drugs and are planning to use them in our experiments) in order to be able to compare ADCs and FDCs in a more adequate way. Secondly, we expect that a different antibody fragment than the scfv and even the minibody molecules used in this work could probably be employed in GD2-directed FDCs more successfully. We currently test other molecules for these purposes. An additional direction for enhancing the efficiency of FDCs may be to optimize their structure to so that to enhance their antigen-binding and internalization into the cells. In general, the creation of a GD2-directed FDC compared to the ADCs is one of our key tasks, which we intend to solve in our subsequent work.

Point 4. In vivo studies of these FDCs would help in improving the scope and impact of the manuscript.

AU: The main goal of this work was to obtain novel GD2-directed constructs based on antibody fragments that would manifest antitumor activity, and to compare them with each other in in vitro models, as well as to determine the feasibility of their use for a wider study. Since we came to the conclusion that the FDCs obtained by us could be optimized in order to increase their cytotoxic properties during the study, it was decided to not conduct in vivo experiments in the current study. Based on our results, we plan to subsequently conduct in vivo experiments comparing ADCs and different optimized FDCs. Even if the optimized FDCs show to be less potent in the in vitro experiments, this disadvantage could be compensated in vivo due to their smaller size that typically results in a better penetration and distribution in solid tumors.

Round 2

Reviewer 1 Report

The authors has taken in account all the comment rendering the comprehension of the study more clear. The revised version of the manuscript is very well and highlights appropriately the scope of the paper.